# Assisting Sustainable Entrepreneurial Activities Through the Analysis of Mobile IT Services' Success and Failure Factors

**Chang Hee Yoon [1], Francis Joseph Costello [2] and Cheong Kim [2,3,\*]**

[1] Graduate School of Technology Management, Sungkyunkwan University, Suwon 16419, Korea; ch100.yoon@g.skku.edu
[2] SKK Business School, Sungkyunkwan University, Seoul 03063, Korea; f.costello@g.skku.edu
[3] Korea Airports Corporation (KAC), Seoul 07505, Korea
\* Correspondence: saga@g.skku.edu

**Abstract:** With information technology (IT) now showing advanced capabilities, many new services are being introduced to consumers of smartphones through the various available app stores. Moreover, the recent proliferation of such services related to information and communications technology has seen a momentous rise. Despite this trend, the ever-changing landscape of mobile IT services is creating a serious problem for businesses who are already experiencing fierce market conditions. Thus, in order to maintain the sustainability of an enterprise, it is necessary to make an adequate analysis of the success and failure factors of IT services in order to create a sustained competitive advantage. Considering 22 real IT service cases based on two platform models (merchant model and two-sided model) and through surveys submitted to 11 experienced entrepreneurs in IT services, we conducted a t-test analysis in order to first assess the success and failure factors of the IT service cases. Next, we performed a logistic regression analysis in order to find underlying relationships of our hypothesized model. The results showed that the participants identified 141 success and 101 failure factors in total with the t-tests, confirming that the distinction between success and failure of each IT service assessed was significant. Next, the results from the logistic regression showed which relationships were the best on the basis of the given platform model. Overall, this study was able to identify the main factors that have an influence on the success and failure of IT services based on two identified platform models. In doing so, this paper can help to inform future IT service entrepreneurs and researchers involved in developing new apps based on IT services by providing a guide to what factors need to be considered before going to market.

**Keywords:** mobile IT service; platform business; success factors; failure factors; sustainable entrepreneurial activities

## 1. Introduction

With most industries moving into industry 4.0, information technology (IT) capabilities have advanced dramatically alongside this industrial change. There is no question that this is having a positive influence on the sustainability of business performance, and companies at the forefront of providing IT-enabled services are seeing the benefits [1]. However, it is not just businesses who are benefiting from this, end users too are being introduced to many new IT-based services, with China and the US now leading the way with the greatest number of information and communications technology (ICT) service exports [2]. These services include consulting, system development, integration, operations, infrastructure, sustainment, and outsourcing. Thus, the concept of IT services also includes new and future services related to ICT [3].

The recent proliferation of mobile IT services, defined as various types of software application services designed to be operated on mobile devices (e.g., smartphones, tablet computers, and smartwatches), has significantly risen. Mobile IT services are found when connecting to a wireless network through any mobile device [4]. Following the mobile revolution that started with the iPhone in 2009, services have exponentially grown. At present, Google and Apple are the two prominent smartphone operating-system (OS) platforms available. Additionally, it was estimated that the number of registered applications was 2.8 million on Google Play and 2.2 million in the Apple App Store in 2017. In line with this growth, as Figure 1 shows, the global Apple App Store sales are expected to increase dramatically from $36.2 B in 2016 to $71.7 B in 2020 [5].

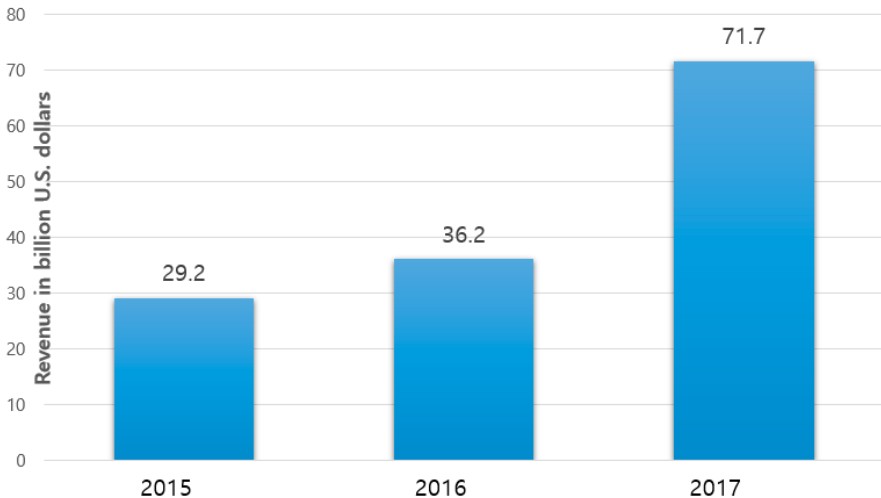

**Figure 1.** App Store revenue estimation [3].

Although promoting IT implementation in a business model has been found to be a key factor for the sustainability of a business's performance within the 4th industrial revolution [1], the number of IT services continuously used by end users is limited. Hence, the survival rate of IT services is constantly decreasing, indicating the need to evaluate factors influencing adoption [6] and resistance [7] of such services.

Nascent entrepreneurs engaged within the IT service sector need to understand the uncertainties of the market. Uncertainties can have an impact upon almost all the different stages of the entrepreneurial process; therefore, understanding the success or failure factors of previous firms can help entrepreneurs deal with uncertainties before acting on an opportunity. Furthermore, assessing the risks surrounding business opportunities is also an essential component in supporting the success of sustainable entrepreneurial activities [8].

In order to help entrepreneurs focus on sustainable entrepreneurial activities (any activity that promotes the sustained growth and advantage relative to competitors [1]), this research has broken the ICT market into four value chains, i.e., content, platform, network, and device, and has further divided the mobile IT service market on the basis of the degree of maturity of a given business platform. These were evaluated using the merchant model and the two-sided model, as Figure 2 shows. The merchant-model-oriented market, geared towards directly servicing the end users, has been built for business continuity. However, several businesses in the ICT domain have combined services through acquisitions and mergers, with a focus on their platform operations. Thus, the world has entered an era of limitless competition with increasingly blurred boundaries [9]. In contrast, the two-sided market model centers on the platform businesses that directly connect users and providers. Due to the ever-changing landscape of mobile IT services, these businesses are experiencing a fierce struggle for existence [10]. Moreover, there are many approaches for entering the market and maintaining successful services, which only adds to this burden [11].

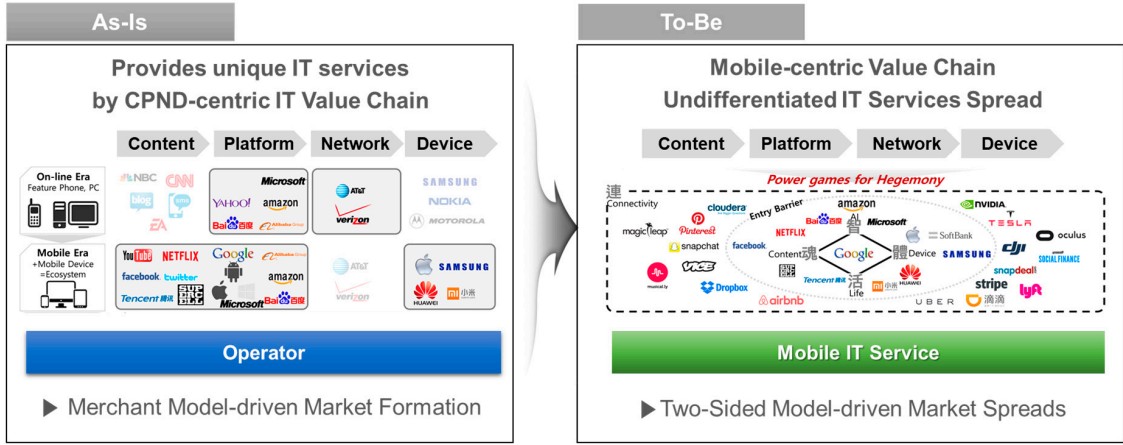

**Figure 2.** Differentiating information and communications technology (ICT) business by competitive flow [9].

Generally, service loyalty increases when users benefit from new services that offer similar, or greater, utility than those of other providers [12]. For services having similar utility levels, there is a tendency to use existing services continuously, especially when switching costs exist or these costs increase [13]. From this, it can be interpreted that new service providers preparing to enter the market should master unique factors to attract users and maintain satisfaction, which ultimately leads to a businesses' success [14]. Success and failure factors are the most fundamental elements for corporations that provide mobile IT services and who want to have a sustainable business model. Especially, for entrepreneurs that set up mobile IT services as their primary business area, these success and failure factors are an extremely critical keystone to sustain their businesses as well as survive within a highly competitive market.

The consequences of this business success are also absolutely relevant to the survival of the company responsible for the service [15]. Maintaining an absolute and sustained lead has proven difficult in light of the recent competition in this market. Unlike traditional business in the past, it is easy for companies that have achieved market dominance through technology to lose the market dominance in a quick manner. Failure to invest in new businesses, inertia of internal organizational activities, and failure to allocate resources are all contributing factors to this recent phenomenon [16]. Thus, in order to maintain the sustainability of an enterprise, one must adequately prepare the business by offering new services [8]. This makes the analysis of the success and failure factors specific to each enterprise a must be priority to create a sustained competitive advantage [17].

Industry 4.0 is no longer a topic only suited for large organizations; it is also an agenda that should be embraced by SMEs (small-to-medium enterprises) as well as individual entrepreneurs [1] who want to create innovative IT services. In this fully fledged mobile age, a study on what supported the successful implementation of the Korean mobile IT services seems fitting. We analyzed the factors of success and failure according to the maturity levels of the respective platform businesses in order to achieve this goal, as well as make recommendations on how these factors can be implemented into sustainable activities that will aid an entrepreneur in their future decisions.

This paper is organized in the following way. Firstly, we describe the definition of IT services on the basis of success and failure factors and explain the rationale behind each as a research hypothesis. Also, we discuss the reasons for choosing mobile services that have been developed in South Korea. The next section shows how the factors for success and failure were collected, including the surveys conducted from experienced IT entrepreneurs. Next, we present the methods of analysis used, including the test of assumptions made for relevance and significance. The last chapter summarizes and draws conclusions on the results of this study and finally presents the limitations and potential future research directions.

## 2. Theoretical Discussion and Hypotheses

### 2.1. Mobile IT Services

Recently, the concept of a sustainable information society has started to take prominence. The concept is based on the use of ICT as a key enabler of sustainability within an organization [18]. However, mobile IT services suffer from the problem of users not using the services for a prolonged time. Defined as sustainable use, this can be simply seen as the idea of using something for a purpose and combining the everyday meaning of the adjective "sustainable" [19]. Thus, the use of mobile IT services should be seen as a key enabler of growth and sustainability for many organizations engaged in creating or adopting IT services [20].

Specific to IT services, their functions to users can be defined as follows: productivity (enhancing work efficiency and practicality of life); entertainment (focusing on user interest and fun, including games, sports, music, photography, and video); information (information provided to users throughout a lifetime); and communication (providing relationships and maintenance through smooth communications among users) [21]. The interplay becomes more granular with the ever-growing and evolving trend of mobile services, subsequently blurring the boundaries further. [22]

More recently, platform services have emerged in such a way that service providers can create new value through brokering needs among providers and their end users. This allows the providers to reduce costs and increase their revenue in an increasingly competitive manner. However, this requires consideration of the diverse needs of the end users, under the provision they receive unilaterally prepared service content [23]. For IT, platform services have resulted in a two-sided market, comprising app developers and app purchasers who focus on mobile applications. This has resulted in the "same-sided effect," which increases utility value as the community in which they belong grows. It has also resulted in the "cross-sided characteristic," which increases utility value as another community grows [24].

Within these trends, platform businesses have recently expanded [25]. A platform business utilizes a business model that mediates the needs of diverse groups during transactions. This has the effect of promoting interactions among members of these groups to create economic market blocs.

This refers to the activities connecting and adjusting the platform centricity of two or more customer groups that are mutually needed for a transaction [26]. For these platform businesses, the enterprise should be responsible for the design, operation, and upgrade of the platform; they should possess differentiated technologies that are difficult to mimic; there should be common interests among platforms, enterprises, and participants; and there should be continuous scalability based on the participants [27,28]. The nature of these platform businesses has led to the tendency for market winners to take all. Hence, having a correct business strategy at the time of market entry has become critical to the success or failure of the platforms [29].

### 2.2. Factors and Criteria for IT Service Success and Failure

Previous research on the success and failure factors in the IT service industry can be broken into two levels of analysis. First, Yoon et al. [30] suggest that for sustainable growth in the IT service industry, success factors should be viewed through an organizational-level lens. They found that collaboration between two companies, including domestic and international partnerships, were key factors in the success of an IT service firm. Griffin and Page [31] instead viewed success and failure factors through a product and service lens, reducing the analysis to a within-an-organization scope. Their study presented metrics by dividing the definitions of success and failure into measures of customer acceptance, financial performance, and product level [31]. Considering 77 products and projects, they argued that it is just as effective to distinguish and to measure products and projects based on a business strategy for market conditions. For example, prospector groups prioritize investments over profit, and their results are measured in terms of future lead ability and rates of return. Analyzer groups quickly analyze and emulate industry leaders to maximize the returns on investment (ROI)

and measure success rates using returns over time. They measure the degree of the market confidence of a firm through measuring their market power and ROI [31].

From a service perspective, research conducted on platform businesses has defined success and failure on the basis of the size of user choices regarding financial and consumer preferences, market share as a marketing criterion, and business ranking in terms of sales and net income [23]. For the success of a platform business, convergence of open business models, differentiation required to lock-in users and developers, and clear definitions of revenue models have been suggested. Furthermore, the success factors of products and services developed and released worldwide differ from country to country. The Mishra research team compared the 288 newest products from South Korea with prior achievements in China and Canada through analyses of the performance factors. They found that similar success factors existed in each country, although none of the factors was common to the entire world [32]. Thus, it was argued that analyses should be performed in each country in order to properly examine the specific success factors.

### 2.2.1. Maturity of the Platform Business

IT services can be divided into merchant models (operators unilaterally provide services to users, similar to traditional market concepts) and two-sided models (a platform divided into two or more groups with different purposes [e.g., suppliers and users]) [29]. Figure 3 describes the differentiating market model based on the maturity of the platform business.

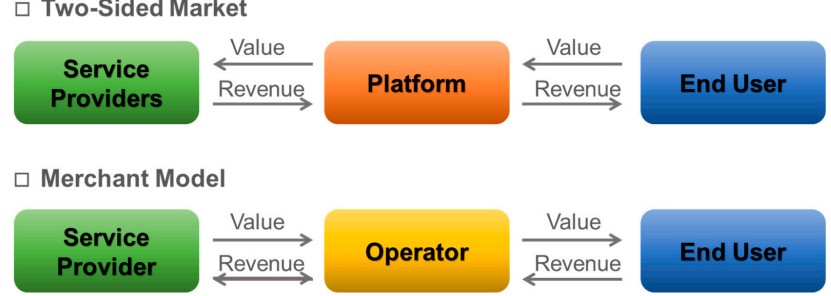

**Figure 3.** Differentiating the market model based on the maturity of the platform business.

The general merchant model is operated in a closed manner without additional stakeholders. It is provided to users by adding an operating revenue to services and content of the first service provider. This allows for the provision of a high-quality and reliable service that unilaterally appeals to users. In contrast, the two-sided model provides users with a range of requirements. Firstly, it takes an interim fee for any transactions while focusing on the platform on which the transaction is made to meet the users' needs [26]. In addition, due to the recent proliferation of mobile platforms, app stores have become common and continue to spread as they are applied to individual merchant model services around mobile E-commerce [33]. Furthermore, if we consider the huge success of the mobile IT business cases whereby a high level of platform maturity exists, such as Uber [34] and Airbnb [35], it is feasible to argue that the maturity of the platform directly affects the success of the mobile IT service. Henceforth, we can postulate a hypothesis concerning the relationship between the level of platform business maturity and the success factors of mobile IT services, as follows:

**Hypothesis 1.** *The success factors of mobile IT services are affected by the level of platform business maturity that connects service providers and users.*

### 2.2.2. Establishment of Interests

Because smartphones are perceived as living necessities, IT services have evolved to provide more personal comfort to users. Many IT services are created and used in the form of applications (apps). However, new services that lack any clear competitive factors compared with competitors already in

the market find that these new apps are usually buried into the depths of the app stores right after launching into the market. [36]. By differentiating an app from existing services based on actual user needs, apps that provide an interesting service are more likely to survive in the market.

The differentiation of IT services is largely driven by suppliers who supply quality services to the platform. For this purpose, the platform operators need to establish meaningful engagement for the suppliers and users in a coherent manner. Value co-creation, in which platform operators, service providers, and users realize mutual benefits has recently been defined as an integral part of the platform business and its wider success [37]. As such, once suppliers and consumers converge on the basis of establishing mutual interests, a sustainable business is possible through the formation of network effects within the platform business [38–41]. Therefore, we propose the following hypothesis in the perspective of establishment of interests:

**Hypothesis 2-1.** *Differentiating from services launched in the existing market has a positive impact on the success of mobile IT services.*

### 2.2.3. Ecosystem Establishment

In the past, the mobile ecosystem within IT services was developed into a mobile ecosystem centered on the OS platforms, including smartphone development [42]. Recently, the scope of service delivery has expanded to include the development of mobile web technologies and cloud services. Related businesses continue to expand and develop into a new mobile ecosystem where user options are enhanced openly without terminals or OS limitations. These developments allow users to enjoy a borderless, homogenous, and seamless experience [43]. One of the successful IT service business cases is Netflix. They provide a borderless service provision with various channels, also known as n-channel, which offers their contents on mobile, personal computer (PC), and even television [44–46]. Thus, building the platform's ecosystem with an emphasis on users' convenience helps in creating a common value through the operator's sustainable business [47], thus we postulate the hypothesis below:

**Hypothesis 2-2.** *Providers offering borderless services through the ecosystem with various channels will have a positive impact on the success of mobile IT services.*

### 2.2.4. Technical Completeness

The technical completeness of IT services is directly linked to usability and is reflected in the Apple App Store via user ratings. Song Chi-hun et al. [48], in their study of mobile applications, stated that online oral information (e.g., consumer ratings registered in the App Store, the number of consumers participating in the App Store, and the number of app developments) was related to the download performance shown in the App store. This is a highly reliable system in which users can base their opinion on technology associations (e.g., satisfaction, drivability, and compatibility) and thus increase the chance of users actively engaging with an App. Furthermore, the utility and convenience of timeliness and quality of service, based on the completion and delivery of the technology, are closely related to mobile service satisfaction and customer loyalty [49]. Especially, when the mobile services deal with sensitive customer information, such as identification numbers, bank account numbers, etc., technical completeness is one of the robust approaches for acquiring customers' satisfaction [50–52]. Hence, we suggest the following hypothesis in the perspective of technical completeness:

**Hypothesis 2-3.** *Technical completeness will have a positive impact on the success of mobile IT services.*

### 2.2.5. Size of the Launching Enterprise

There are different approaches used by venture companies and large companies for launching services in a rapidly changing IT service business environment. This is because effective strategies vary depending on the size of the entity, essentially acting as important variables in performance [53].

Venture companies are capable of coping with the rapidly changing market environment, but they cannot develop, source, or manage after-sales services by themselves [54]. In contrast, large enterprises often fail to enter a new market because of their focus on fund investment and development, services/content sourcing through brand power, and maintaining stability. Thus, they fail to respond proactively to the rapidly changing market environment [55]. Thus, the hypothesis below suggests the relationship between the size of a launching enterprise and the success of mobile IT services:

**Hypothesis 3-1.** *The development and rollout of mobile IT services by small and medium venture firms have a negative impact on success.*

### 2.2.6. PC Interconnection

As we enter a new era in the mobile ecosystem, the boundaries between wired and wireless is slowly disappearing around IT services. With this, mobile service applications continue to evolve into a more tightly coupled platform using cloud and web services [56]. In the past, applications for mobile devices were divided into native and web apps. Native apps took the form of indigenous applications on a device. This has the advantages of quick download from an app store and usability within the device's resources. Native apps generally depend on the device and the OS platform and have the disadvantage of requiring individual applications to spread business. In contrast, web apps offer usability with a variety of devices and OS platforms which are based on a web service to support the service-orientated business. However, application businesses via an app store in the form of web pages are not possible. Recently, hybrid apps have emerged to support multi-OS platforms in the form of applications, enabling full-scale service business expansion [57]. In Korea, it is still too early to completely replace web services with mobile apps, mostly because they are restricted within an environment based on ActiveX [58]; thus, we propose the following hypothesis:

**Hypothesis 3-2.** *Providing expanded services in conjunction with PCs will have a positive impact on the success of mobile IT services.*

### 2.2.7. Consumer Pricing Strategies

IT services can be divided into free, paid, and partially paid business models [59]. For the free pricing strategy-based model, advertising is adopted primarily through information-enabled services, and thus users are not charged. However, the operating and development costs are reimbursed by the advertising service provider after adopting an advertising model for service operations. In contrast, the paid pricing strategy-based model can be adopted and means costs are reimbursed directly by users. This can be through lump sums or regular payments during a given period when the service is delivered. The service provider can obtain users with a high reputation for services and content or a large number of loyal users. In recent years, the preferred pricing strategy has seen the partially charged model, or "freemium" model, take prominence. This model provides additional services and functions while eliminating advertising via in-app payments. Users can therefore use the services without attaching themselves to regular payments, subscriptions, or paid usage periods [60]. Henceforth, we propose a hypothesis concerning the relationship between consumer pricing strategies and the success of mobile IT services, as follows:

**Hypothesis 3-3.** *Paid services for users will have a negative impact on the success of mobile IT services.*

### 2.2.8. Innovations in New Technology

In regard to mobile services, new concepts of IT services based on innovative technologies continue to be developed and introduced into the marketplace. In a study by Kleinschmidt and Cooper [61], the relationship between innovation and performance of new products was analyzed through segmenting the level of innovation of new products into three types of innovation: high,

mid-level, and low. They reported that, for highly innovative (e.g., first-mover) products, the risks of failure in the market were relatively low because of higher-than-expected ROI. For intermediate-level (e.g., fast-follower) innovations, the risk of failure was reduced. However, significant performance was difficult to achieve because of lower-than-expected synergies among marketing, technology, and production. Therefore, we propose the following hypothesis in the perspective of innovations in new technology:

**Hypothesis 3-4.** *Leading the market by introducing new technologies with a first mover propensity will have a positive impact on the success of mobile IT services.*

## 3. Research Methodology

### 3.1. Research Model

On the basis of previous research, we now discuss the definitions of success and failure and how to measure them. The number and variety of recent IT services provide many cases where success and failure can be clearly determined, whereas it is difficult to make individual assessments uniformly. Therefore, when determining the success or failure of IT services through participants using a questionnaire, the constructs were measured considering the mean and using t-tests to determine any significant difference between the groups investigated within this study. Three subjective evaluation factors (i.e., interests established by the research hypothesis, ecosystem establishment, and technology completeness) for merchant and two-sided models, which are separated from service and platform businesses, were applied to success and failure analyses via logistic regression of four objective evaluation factors (i.e., scale of companies examined separately, PC interworking, consumer billing, and introduction of new technologies). Our research model is presented in Figure 4.

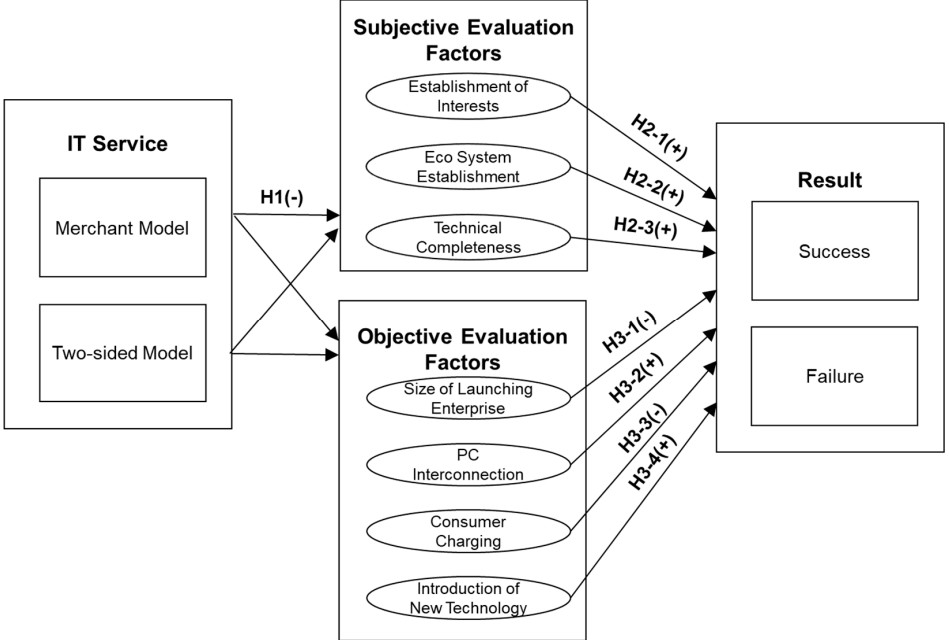

**Figure 4.** Research model.

### 3.2. IT Service Case Selection

For this study, a total of 98 cases of ICT convergence were obtained from 65 ICT experts from 17 February to 18 February, 2018. Of these 98 cases, 46 were cases of success, and 52 were cases of failure. These ICT convergence cases were re-classified as global and regional releases and were selected as original Korean IT services. The final 22 IT service cases, as Table 1 shows, were selected

by removing services that were too old to be independently verified and whereby system units were difficult to evaluate individually for success and failure factors (see Appendices A and D for detailed information of the 22 IT service cases).

Furthermore, although some of the services within the sample were well known in the market via advertising and media coverage, in some cases, the services were unfamiliar due to the fact they had originated from start-ups. Additionally, in attempting to choose appropriate cases for analysis, the final selection of services was not performed on a purely market recognition basis. Instead, the services were chosen on the basis of the fact they were established through an application, with some in the process of expanding their service functionality through PC-based platforms.

**Table 1.** Final list of analyzed service cases.

| Type | Name | Enterprise | Category | Index |
|---|---|---|---|---|
| **Merchant Model** | T Map | SK Telecom | Telecommunication | Information |
| | Golfzon | Golfzon | Leisure | Entertainment |
| | Naver Camera | Naver | Life | Productivity |
| | Moneta Card | SK Telecom | Finance | Productivity |
| | Bank Wallet | Bank union | Finance | Productivity |
| | Bluelink | Hyundai Motors | Maintenance | Productivity |
| | Beat | Beat | Music | Entertainment |
| | Socar | Socar | Transportation | Productivity |
| | Jeongyookgak | Jeongyookgak | Shopping | Information |
| | Kakao Bank | Kakao | Finance | Productivity |
| | T Money | KSCC | Finance | Productivity |
| **Two-Sided Model** | Gifticon | SK Telecom | Shopping | Productivity |
| | Market Kurly | Kurly | Shopping | Information |
| | Baedal Minjok | Woowa Brothers | Life | Information |
| | Samsung Pay | Samsung Electronics | Finance | Productivity |
| | Syrup | SK Planet | Shopping | Productivity |
| | Wibee Talk | Woori Bank | Finance | Productivity |
| | The Pirates | The Pirates | Shopping | Information |
| | Kakao Taxi | Kakao | Transportation | Information |
| | Coocha | Coocha | Shopping | Information |
| | Coupang | Coupang | Shopping | Information |
| | Hwahae | Birdview | Beauty | Information |

*3.3. Survey to IT Entrepreneurs*

A survey was submitted to entrepreneurs who had experience in similar fields of this study and who showed an interest in creating their own IT start-up. To assist participants who were lacking awareness of IT services, the survey introduced each service and their assessments of success and failure. By displaying these descriptions of the IT services, we were able to increase the understanding by introducing a detailed overview of the business models and operating methods for each service selected. In addition, we introduced detailed cases, demos, and competitive services to participants when they were unfamiliar with the presented service. Individual surveys conducted simultaneously led to a seven-point evaluation of a total of three categories: establishing interests selected as factors for success and failure of mobile IT services for this study; establishing ecosystems; and completing technologies. A pre-arranged questionnaire containing survey items separated into each service was distributed to the participants. The survey participants' successes and failures were subsequently judged, and, with the survey items, the participants were asked to list factors of success and failure for the services they chose. When introducing each service, the results of separate judgments about success and failure were not considered, so that the actual operational results would not be reflected in the survey results of the participants.

## 4. Empirical Analysis

### 4.1. Survey

A total of 22 IT service cases originating in South Korea were selected to analyze the success and failure factors of IT services. Next, we invited 11 people with high entrepreneurial skills and experience in this industry who showed an interest in creating or being a part of an IT venture start-up into our survey [62]. The average age and work experience of the participants were 37 and 10 years, with all survey participants having experience of working for companies that have more than 300 employees. The specific industries in which previous experience was obtained were represented by four business-planning workers, two consulting workers, and works with various other jobs, including education, marketing, analysis, operation, and finance/accounting [63]. The demographic composition of the participants is shown in Table 2.

**Table 2.** Survey participants.

| No. | Age | Profession | Career (years) | Location |
|-----|-----|------------|----------------|----------|
| 1 | 49 | Consulting | 20 | Seoul |
| 2 | 40 | Consulting | 18 | Seoul |
| 3 | 35 | Education | 8 | Seoul |
| 4 | 39 | Operation | 6 | Seoul |
| 5 | 35 | Planning | 9 | Seoul |
| 6 | 36 | Analysis | 8 | Seoul |
| 7 | 37 | Planning | 7 | Seoul |
| 8 | 31 | Planning | 7 | Gyeonggi |
| 9 | 35 | Finance | 9 | Seoul |
| 10 | 35 | Planning | 3 | Gyeonggi |
| 11 | 37 | Marketing | 11 | Seoul |

First of all, we developed three subjective constructs (establishment of interests, ecosystem establishment, and technical completeness) to be analyzed in the survey. These constructs were adopted on the basis of the fact that these factors have been analyzed in former research. The questionnaire in this study was answered on a seven-point Likert scale that ranged from "strongly disagree (1)" to "strongly agree (7)." The questionnaire and sources are shown in Table 3 (see full table of the questionnaire in Appendix B). Note that the other four objective constructs will be presented later in Section 4.3.

**Table 3.** Summary of the questionnaire.

| Questionnaire | Source |
|---------------|--------|
| Establishment of Interests | Kim et al. [36], Smedlund [37], Parker and Van Alstyne [38], West [39], Boudreau [40], Parker and Van Alstyne [41] |
| Ecosystem Establishment | Iansiti and Levien [42], Kim and Lee [43], Bell and Koren [44], Jenner [45], Narayanan and Shmatikov [46] |
| Technical Completeness | Song et al. [48], Jung and Lee [49], Chung and Kwon [50], Al-Jabri and Sohail [51], Lin [52] |

The survey was conducted to understand the participants' subjective evaluation of what constitutes success and failure, establishment of interests, ecosystem establishment, and technology completeness. The objective evaluation criteria, such as maturity of the platform, size of the launching enterprise, PC interconnection, consumer charging, and introduction of new technology were evaluated based on the online information given by government statistics [64], press releases, and firms' official websites. SPSS Statistics 2.3 (IBM, New York, NY, USA) was used to then verify the statistical significance of the assessments made.

## 4.2. Results of the t-Test

The subjective factors for each success and failure in the 22 IT services were assessed using t-tests (see full table of the objective constructs in Appendix C). Of the total 22 services analyzed, 11 survey participants identified 141 success and 101 failure factors in total. First, the regularity of the continuous variables was verified by analyzing the descriptive statistics of the variables by their category. As a result of establishment of interest, the mean was 4.541, and the standard deviation was 1.7759. For ecosystem establishment, the average was 4.533, and the standard deviation was 1.8085. for technical completeness, the average was 4.744, and the standard deviation was 1.6575. In addition, a test of normal distribution was conducted through the measurement of skewness and kurtosis, and both results were found to be valid. The result of the descriptive statics is presented in Table 4.

**Table 4.** Descriptive statistics.

| Category | N | Min. | Max. | Mean | SD | Skewness | Kurtosis |
|---|---|---|---|---|---|---|---|
| Establishment of Interests | 242 | 1 | 7 | 4.541 | 1.7759 | −0.154 | −1.04 |
| Ecosystem Establishment | 242 | 1 | 7 | 4.533 | 1.8085 | −0.16 | −1.115 |
| Technical Completeness | 242 | 1 | 7 | 4.744 | 1.6575 | −0.286 | −0.825 |

In order to verify the results of the *t*-test from our research, we adopted the assumption of variance in the population. Welch's *t*-test was conducted for services based on the maturity of the platform. In order to retain statistical robustness, we also used 5000 bootstrapped samples to verify our test statistics and *p* values. For each success and failure case, the Welch's *t*-values for each factor were the following: for establishment of interests, the value was −12.746, for ecosystem establishment, it was −19.049, and for technical completeness, it was −10.960. These numbers were valid for both distinctions between success and failure. The result of the Welch's *t*-test for service total are presented in Table 5.

**Table 5.** Result of Welch's *t*-test: service total.

| Category | Mean | | SD | | F |
|---|---|---|---|---|---|
| | Success | Failure | Success | Failure | |
| Establishment of Interests | 5.504 | 3.198 | 1.3073 | 1.4424 | −12.746 * |
| Ecosystem Establishment | 5.723 | 2.871 | 1.1026 | 1.1804 | −19.049 * |
| Technical Completeness | 5.574 | 3.584 | 1.1846 | 1.5249 | −10.960 * |

$* p < 0.001$.

Of the total cases, the survey participants determined that the merchant model had 70 successes and 51 failures. For all the three subjective evaluation factors, the t-values for success and failure were validated. The Welch's *t*-value was −9.435 for establishment of interests, −13.084 for ecosystem establishment, and −6.420 for technical completeness. The magnitude of the differences between success and failure among the given factors was analyzed in the order of ecosystem building, stake-making relationships, and technology completeness. The result of the Welch's *t*-test for the merchant model is shown in Table 6.

**Table 6.** Results of Welch's *t*-test: merchant model.

| Category | Mean | | SD | | F |
|---|---|---|---|---|---|
| | Success | Failure | Success | Failure | |
| Establishment of Interests | 5.514 | 3.020 | 1.3593 | 1.4898 | −9.435 * |
| Ecosystem Establishment | 5.700 | 2.784 | 1.1715 | 1.2380 | −13.084 * |
| Technical Completeness | 5.543 | 3.765 | 1.2294 | 1.6922 | −6.420 * |

$* p < 0.001$.

Of the total cases, the survey participants determined that the two-sided model had 71 successes and 50 failures. Differences in success and failure factors were identified in the order of establishment of interests, ecosystem establishment, and technical completeness. The Welch's t-values were analyzed to the effective level, obtaining −8.572 for establishment of interests, −13.854 for ecosystem establishment, and −9.438 for technical completeness. The results of the Welch's t-test for the two-sided model are shown in Table 7.

**Table 7.** Results of Welch's t-test: two-sided model.

| Category | Mean | | SD | | $F$ |
|---|---|---|---|---|---|
| | **Success** | **Failure** | **Success** | **Failure** | |
| Establishment of Interests | 5.493 | 3.380 | 1.2635 | 1.3834 | −8.572 * |
| Ecosystem Establishment | 5.746 | 2.960 | 1.0381 | 1.1241 | −13.854 * |
| Technical Completeness | 5.606 | 3.400 | 1.1769 | 1.3248 | −9.438 * |

* $p < 0.001$.

### 4.3. Results of Logistic Regression

In addition to the three constructs investigated through the survey, four objective evaluation factors were added as dummy variables, including size of launching enterprise, PC interconnection, consumer charging, and introduction of new technology. We obtained these dummy variables by online research using keywords (the names of the service and the construct, e.g., T map and PC interconnection). If there was corresponding information, the dummy variable 1 was given, otherwise the dummy variable 0 was given. Note that the sizes of the corporations were set according to the categorization by the Korean government statistics [64]. These constructs were used to analyze the correlation among variables and to identify the relationship between success and failure, as shown in Table 8.

In addition, logistic regression was conducted in order to discover the impact of factors on the success and failure of mobile IT services. The results were verified with likelihood ratio and Exp (B) function and shown to be valid as follow.

First, our results showed that building an ecosystem was important for all 22 services (i.e., merchant model and two-sided model). This translates into users considering the usability of associated services beyond their basic use when using IT services. This suggests that building ecosystems should be considered before planning and developing any new IT service in the future [65]. In the service-wide analysis (two-sided or merchant), the maturity of the platform business of IT services was added as a dummy variable to identify the impact of success and failure. Among the factors, ecosystem deployment and technology completion were found to be significant for success and failure. The results of the logistic regression analysis for service total are shown in Table 9.

**Table 8.** Correlation analysis.

| | Category | 1 | 2 | 3 | 4 | 5 | 6 | 7 | 8 | 9 | 10 |
|---|---|---|---|---|---|---|---|---|---|---|---|
| | **1. Success Status** | 1 | | | | | | | | | |
| | **2. Platform Status** | 0.008 | 1 | | | | | | | | |
| **Subjective** | **3. Establishment of Interests** | 0.642 ** | 0.044 | 1 | | | | | | | |
| **Evaluation** | **4. Ecosystem Establishment** | 0.779 ** | 0.034 | 0.795 ** | 1 | | | | | | |
| **Factors** | **5. Technical Completeness** | 0.593 ** | −0.030 | 0.601 ** | 0.663 ** | 1 | | | | | |
| | **6. In-App Purchase** | −0.253 ** | 0.236 ** | −0.204 ** | −0.305 ** | −0.212 ** | 1 | | | | |
| **Objective** | **7. Payment Paid** | 0.002 | −0.183 ** | 0.012 | 0.044 | −0.091 | −0.516 ** | 1 | | | |
| **Evaluation** | **8. PC Interconnection** | 0.109 | 0.273 ** | 0.054 | 0.089 | −0.040 | 0.236 ** | 0.000 | 1 | | |
| **Factors** | **9. Venture Status** | −0.150 * | −0.183 ** | −0.044 | −0.121 | 0.014 | −0.280 ** | 0.083 | −0.548 ** | 1 | |
| | **10. New Technology Leaderships** | 0.267 ** | 0.488 ** | 0.219 ** | 0.271 ** | 0.141 * | −0.184 ** | −0.036 | −0.098 | −0.232 ** | 1 |

$** p < 0.05, * p < 0.1$.

**Table 9.** Results of Logistic Regression: service total.

| Category | B | S.E. | Wald | P | Exp(B) | 95.0% C.I. for EXP(B) | |
| --- | --- | --- | --- | --- | --- | --- | --- |
| | | | | | | Lower | Upper |
| Ecosystem Establishment | 1.802 | 0.274 | 43.293 | 0.000 *** | 6.059 | 3.543 | 10.362 |
| Technical Completeness | 0.703 | 0.243 | 8.377 | 0.004 *** | 2.02 | 1.255 | 3.252 |
| Constant | 10.622 | 1.693 | 39.375 | 0 | 0 | | |

*** $p < 0.01$

General merchant models are significant for establishing an ecosystem, whether technical competition or connecting through a PC or a new technology leadership. For the merchant model, when usability is concerned, we interpreted that a new technology leadership has a greater chance to succeed when there is more active use. Primarily this would be achieved by providing scalability, such as use through a PC as well as other mobile phone platforms. The results of logistic regression for the merchant model re shown in Table 10.

**Table 10.** Result of logistic regression: merchant model.

| Category | B | S.E. | Wald | P | Exp(B) | 95.0% C.I. for EXP(B) | |
| --- | --- | --- | --- | --- | --- | --- | --- |
| | | | | | | Lower | Upper |
| Ecosystem Establishment | 1.619 | 0.429 | 14.259 | 0.000 *** | 5.05 | 2.179 | 11.705 |
| Technical Completeness | 0.762 | 0.371 | 4.227 | 0.040 ** | 2.143 | 1.036 | 4.434 |
| PC Interconnection | 3.756 | 1.171 | 10.281 | 0.001 *** | 42.773 | 4.306 | 424.863 |
| New Technology Leadership | 1.908 | 0.903 | 4.465 | 0.035 ** | 6.738 | 1.148 | 39.546 |
| Constant | −12.042 | 2.835 | 18.048 | 0 | 0 | | |

*** $p < 0.01$, ** $p < 0.05$.

In contrast, in the IT platform service business, the two-sided model has a greater maturity within the marketplace. Hence, it differs from the typical merchant model in terms of its success and failure factors. On the basis of the analysis, the identified success and failure factors were seen within establishment of ecosystems and completion of technologies. In addition, the analysis allowed us to infer that it is possible to succeed with a two-sided model if it is differentiated from other services. Specifically, differentiation must occur against services that were released in the past and against those still existing in the current marketplace. Furthermore, the potential for success is reduced when users feel uncomfortable about the usability of a service, usually due to the incomplete nature of a technology in comparison to that of an available one. This means that preparing a mobile IT service business based on the two-sided model will require a new service that is more attractive than the existing service on offer. Coupled with this, a complete technical solution, one that is available across multiple ecosystems and has enhanced usability, will help it to survive in the marketplace. The results of logistic regression for the two-sided model are shown in Table 11.

**Table 11.** Result of logistic regression: two-sided model.

| Category | B | S.E. | Wald | P | Exp(B) | 95.0% C.I. for EXP(B) | |
| --- | --- | --- | --- | --- | --- | --- | --- |
| | | | | | | Lower | Upper |
| Ecosystem Establishment | 1.934 | 0.431 | 20.11 | 0.000 *** | 6.919 | 2.971 | 16.114 |
| Technical Completeness | 0.935 | 0.426 | 4.821 | 0.028 ** | 2.546 | 1.106 | 5.864 |
| Constant | −12.328 | 2.829 | 18.996 | 0.000 *** | 0 | | |

*** $p < 0.01$, ** $p < 0.05$.

*4.4. Hypothesis Verification Results*

Logistical regression also helped to identify the impact of these factors on the success and failure of IT services.

As shown in Tables 9–11, for hypothesis 1, based on the platform propensity, there is only a difference between success and failure factors, and the impact of maturity itself was not accepted. As a result, mobile IT services need to focus on key elements rather than maturity of the platform business. Regarding hypothesis 2-1, it was rejected at all levels of the platform maturity, and it was shown that establishing a new relationship that is different from existing services cannot be considered to affect the success of an IT service. Therefore, this shows that entrepreneurs need to create a business that existing stakeholders can familiarize themselves with. For the implementation of the ecosystem identified in hypothesis 2-2, the results verified that this aspect of an IT service business platform should be considered as one of the most important factors. Establishing and adopting this within the overall service model is a must, once an appropriate service design has been identified. For technology completeness also identified in hypothesis 2-3, it was also shown to be an important factor as it was adopted at all maturity levels. This means that mobile IT services are not able to succeed if users feel uncomfortable with their use, and full functional verification is required when developing them. Hypothesis 3-1, which identified success and failure factors of services released by a venture company, was only adopted for general merchants. These were generally entrepreneurs who found it difficult to start a venture or start-up, for which initial business preparation was relatively difficult, and the quantity and quality of the content provided was important. The PC interconnection identified in Hypothesis 3-2 was adopted only for the merchant model, which proved important for it to be more usable and ready for market. This provided more frequent use by users of the service in more varied environments. Hypotheses 3-3, which confirmed consumer willingness to pay, and hypotheses 3-4, which confirmed first mover propensity, were both rejected, confirming that there was little linkage between these factors and success and failure factors. Table 12 summarizes the comprehensive results for hypotheses verification.

## 5. Discussion

### 5.1. Conclusions and Implications

In a time of industry 4.0, the sustainability of business performance is directly linked to the IT capabilities one has, with IT-enabled services creating a sustained advantage for organizations [1]. This trend has been found in countries like China, the US, Japan, and South Korea, all leading the way in ICT service exports [2]. However, with a saturated market and many uncertainties in IT service start-ups [8], nascent entrepreneurs engaged within the IT service sector need to better understand the success and failure factors involved.

Therefore, this study attempted to analyze the success and failure factors of IT service planning and development within the smartphone application market. In doing so, we attempted to identify and suggest the main factors that have an influence on the success and failure of IT services. This will help to inform future IT service entrepreneurs in how to develop new apps that have a sustained competitive advantage for their businesses. In order to test our hypotheses, we analyzed 22 mobile IT-related services that were started in South Korea as either a merchant or a two-sided model. This study also reflected upon the level of maturity of the platform business between a service provider and a user as a way to further analyze any variations between immature and mature platforms. We conducted a questionnaire survey based on three factors and analyzed the significance of success and failure using a t-test. We also analyzed the in-app settlement, paid settlement, PC integration, venture development, and new technology leadership. Additionally, logistic regression analysis and the likelihood ratio Exp (B) function was applied to the nine factors in order to test the constructs. A schematic diagram of this process can be seen in Figure 5.

**Table 12.** Hypothesis verification results.

| | Hypothesis | Keyword | Service Total | Merchant Model | Two-Sided Model |
|---|---|---|---|---|---|
| *H1* | The success factors of mobile IT services are affected by the level of maturity of the platform business that connects service providers and users. | Maturity of the Platform Business | Reject | - | - |
| *H2-1* | Differentiating from services launched in the existing market has a positive impact on the success of mobile IT services. | Establishment of Interests | Reject | Reject | Reject |
| *H2-2* | Providers offering borderless services through the ecosystem with various channels will have a positive impact on the success of mobile IT services. | Ecosystem Establishment | Accept | Accept | Accept |
| *H2-3* | Technical completeness will have a positive impact on the success of mobile IT services. | Technical Completeness | Accept | Accept | Accept |
| *H3-1* | The development and rollout of mobile IT services by small and medium venture firms have a negative impact on success. | Venture Launching | Reject | Accept | Reject |
| *H3-2* | Providing expanded services in conjunction with PC will have a positive impact on the success of mobile IT services. | PC Interconnection | Reject | Accept | Reject |
| *H3-3* | Paid services for users will have a negative impact on the success of mobile IT services. | Consumer Charging | Reject | Reject | Reject |
| *H3-4* | Leading the market by introducing new technologies with a first mover propensity will have a positive impact on the success of mobile IT services. | New Technology Leadership | Reject | Reject | Reject |

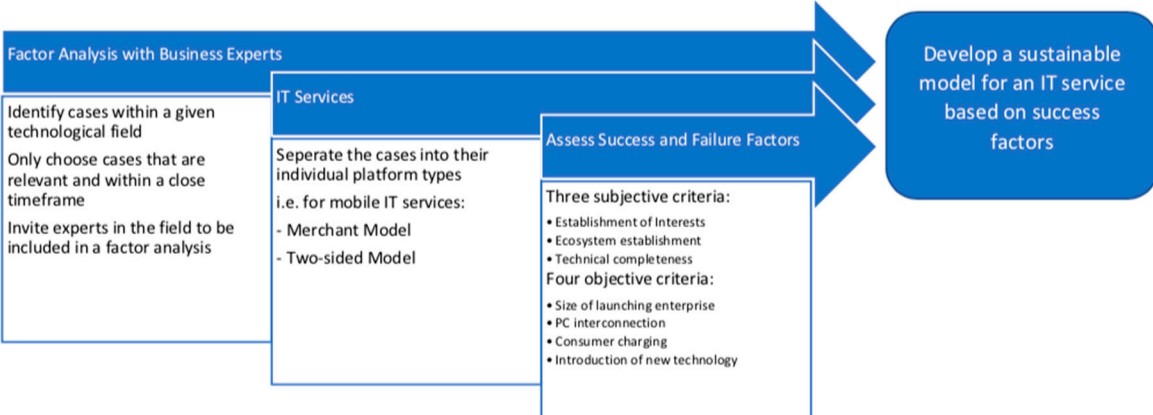

**Figure 5.** Schematic diagram of the action algorithm.

Looking at the findings, ecosystem establishment and technical completeness appeared to be the only two variables that were fully supported by all three tested models. Furthermore, these two variables were the only accepted constructs in terms of the total IT service ecosystem. Despite our original design failing to find significant results for all the hypothesis, we can recommend these two constructs as focal points for any entrepreneur looking to implement an IT service which can be sustained through its start-up phase.

From this study, we propose the following suggestions about IT services preparing to launch in the marketplace. First, we conducted an empirical analysis on mobile-based IT services launched in Korea. Previous studies focused on analyzing factors using case studies that could be evaluated by clear criteria, such as finance/accounting and monthly active users [31]. This study focused on factor analysis and business experts who were actually preparing for a start-up using mobile IT services.

Second, we analyzed the success and failure factors of mobile IT services in the Korean market. When planning an IT service, service design should be made after pre-sorting the factors according to the degree of maturity in the platform business. Here, the success and failure factors of the merchant

and two-sided models, classified according to the degree of maturity of the platform businesses, need to focus on key factors such as ecosystem establishment and technical completeness, which is essential for both sides [32].

Third, this study acts as a guide for planning the preparation of a mobile IT service in the smartphone application market. Users are focusing on perfect usability rather than rejection of paid billing when using IT services, and suppliers should suggest a friendly business model for success. This is because in the case of a new business model, it is difficult for a starter to succeed because of the strong influence of the suppliers who succeeded earlier. In the case of a new business model, it is difficult for the late runners to succeed because of the strong influence of the previous suppliers. This is related to the fact that platform businesses have a strong tendency to win [30]. Therefore, rather than entering into the market using similar existing services, a creative approach to business in a new field is needed based on usability and through absolute technology completeness.

The results from this paper show an agreement with prior research whereby the leading services were shown to be dominant within the IT service industry [29]. This once again highlights the need for newcomers to correctly select the factors that will help to bring success as well as a sustainable model. However, as demonstrated by Griffin and Page [31], identifying effective factors for success and failure is a challenging task, and the results of this study concur with these findings. Despite this paper moving the point of analysis away from case studies to an empirical one, future research will need to further refine ideas for evaluating the IT service industry.

### 5.2. Limitations and Future Recommendations

This study has the following limitations as well as suggests future direction for other researchers and practitioners engaged in similar research and activities.

First, this research is lacking a sizable number of cases. Furthermore, the cases analyzed were more specific to the success and failure of ICT convergence, rather than of mobile IT services. Therefore, it will be necessary to increase the number of specific cases analyzed as well as expand the scope of the analysis with a greater focus on services. Second, we conducted this study by subjectively classifying the successes and failures in the questionnaires for each case. Thus, there were different judgments on successes and failures for the same case. Although the researchers conducted the correlation and significance analysis of the success and failure factors based on the questionnaire, it is now necessary to analyze the influence of each factor by conducting research based on an objective definition of success and failure beforehand.

Third, there is a need for a more in-depth understanding of the cases analyzed by the participants. Most participants in the questionnaire were aware of the service as the subject of the analysis. However, they lacked experience or, in most cases, hands-on experience with the service. Despite making brief introductions to the participants about the services, it cannot be guaranteed that all the participants had the same information about the service as if they actually had hands-on experience. As a result, it was challenging for some of them to make a detailed evaluation of each service. In the future, research should conduct surveys on services that users already have prior experience with or provide enough time for participants to experience the service in advance before undertaking any survey.

**Author Contributions:** Conceptualization, C.H.Y. and C.K.; Data curation, C.H.Y. and F.J.C.; Formal analysis, C.H.Y.; Investigation, C.H.Y., F.J.C., and C.K.; Methodology, C.H.Y. and C.K.; Software, C.H.Y., and C.K.; Validation, F.J.C., and C.K.; Visualization, C.H.Y.; Writing—original draft, C.H.Y., F.J.C., and C.K.; Writing—review & editing, F.J.C. and C.K.

**Funding:** This research received no external funding.

**Conflicts of Interest:** The authors declare no conflict of interest.

## Appendix A

**Table A1.** Analyzed services.

| Name | Model | Enterprise | Category | Launching | General Outline |
|---|---|---|---|---|---|
| T Map | Merchant | SK Telecom | Information | 2002 | Mobile Navigator |
| Gifticon | Two-Sided | SK Telecom | Shopping | 2014 | Real-trade Commerce Solutions |
| Naver Camera | Merchant | Naver | Life | 2015 | Smart Camera |
| Market Kurly | Two-Sided | Kurly | Shopping | 2016 | Premium Food Shopping mall |
| Moneta Card | Merchant | SK Telecom | Finance | 2011 | First Mobile Care |
| Baedal Minjok | Two-Sided | Woowa Brothers | Life | 2010 | Delivery Relay Service |
| Bank Wallet | Merchant | KHTC | Finance | 2013 | Mobile Electric Wallet |
| Blue link | Merchant | Hyundai Motors | Maintenance | 2011 | Connected Car Service |
| Beat | Merchant | Beat | Music | 2014 | Free Music Streaming Service |
| Samsung Pay | Two-Sided | Samsung electronics | Finance | 2015 | Mobile Payment |
| Golfzon | Merchant | Golfzon | Leisure | 2014 | Golf Simulator |
| Syrup Order | Two-Sided | SK Planet | Shopping | 2014 | O2O Pre-order and Pick-up |
| Socar | Merchant | Socar | Transportation | 2011 | Car Sharing |
| Wibitalk | Two-Sided | Woori Bank | Finance | 2016 | Mobile Finance Platform |
| The Pirates | Two-Sided | The Pirates | Shopping | 2013 | Fish Market Linked Market Service |
| Jeongyookgak | Merchant | Jeongyookgak | Shopping | 2016 | Livestock Distribution Start-up |
| Kakao Bank | Merchant | Kakao | Finance | 2016 | Online Bank |
| Kakao Taxi | Two-Sided | Kakao | Transportation | 2015 | Mobile call Taxi Connection Service |
| Coocha | Two-Sided | Coocha | Shopping | 2015 | Hot Deal Comparing Shopping Platform |
| Coupang | Two-Sided | Coupang | Shopping | 2013 | Distribution Specialized e-Commerce |
| T Money | Merchant | KSCC | Finance | 2003 | O2O Payment Platform |
| Hwahae | Two-Sided | Birdview | Beauty | 2013 | Cosmetic Information Service |

## Appendix B

**Table A2.** Full table of the questionnaire.

| Service | Category | Establishment of Interest | Eco System Establishment | Technical Completeness | Success/Failure |
|---|---|---|---|---|---|
| | | 1 (Strongly Disagree) to 7 (Strongly Agree) | | | |
| T Map | Information | | | | |
| Gifticon | Shopping | | | | |
| Naver Camera | Life | | | | |
| Market Kurly | Shopping | | | | |
| Moneta Card | Finance | | | | |
| Baedal Minjok | Life | | | | |
| Bank Wallet | Finance | | | | |
| Blue link | Maintenance | | | | |
| Beat | Music | | | | |
| Samsung Pay | Finance | | | | |
| Golfzon | Leisure | | | | |
| Syrup Order | Shopping | | | | |
| Socar | Transportation | | | | |
| Wibeetalk | Finance | | | | |
| The Pirates | Shopping | | | | |
| Jeongyookgak | Shopping | | | | |
| Kakao Bank | Finance | | | | |
| Kakao Taxi | Transportation | | | | |
| Coocha | Shopping | | | | |
| Coupang | Information | | | | |
| T Money | Shopping | | | | |
| Hwahae | Beauty | | | | |

## Appendix C

**Table A3.** Full table of the objective constructs.

| Service | Size of Launching Enterprise | PC Interconnection | Consumer Charging | Introduction of New Technology |
|---|---|---|---|---|
| | 0 (Small) or 1 (Big) | 0 (No) or 1 (Yes) | 0 (No) or 1 (Yes) | 0 (No) or 1 (Yes) |
| T Map | 1 | 0 | 0 | 1 |
| Gifticon | 1 | 1 | 0 | 1 |
| Naver Camera | 1 | 0 | 0 | 0 |
| Market Kurly | 0 | 1 | 0 | 1 |
| Moneta Card | 1 | 0 | 0 | 1 |
| Baedal Minjok | 0 | 1 | 0 | 1 |
| Bank Wallet | 1 | 0 | 0 | 0 |
| Blue link | 1 | 0 | 0 | 0 |
| Beat | 0 | 1 | 1 | 0 |
| Samsung Pay | 1 | 0 | 0 | 1 |
| Golfzon | 0 | 0 | 0 | 1 |
| Syrup Order | 1 | 0 | 0 | 1 |
| Socar | 0 | 1 | 0 | 0 |
| Wibeetalk | 1 | 1 | 1 | 0 |
| The Pirates | 0 | 1 | 0 | 1 |
| Jeongyookgak | 0 | 1 | 0 | 1 |
| Kakao Bank | 1 | 0 | 0 | 1 |
| Kakao Taxi | 1 | 0 | 0 | 1 |
| Coocha | 0 | 1 | 1 | 1 |
| Coupang | 0 | 1 | 0 | 1 |
| T Money | 1 | 1 | 0 | 0 |
| Hwahae | 0 | 0 | 1 | 1 |

## Appendix D

**Table A4.** Websites of the services.

| Name | Enterprise | URL | Remark |
|---|---|---|---|
| T Map | SK Telecom | https://www.tmap.co.kr/ | |
| Gifticon | SK Telecom | http://www.gifticon.com/ | |
| Naver Camera | Naver | http://camera.line.me/ | Change to Line Camera in 2013 |
| Market Kurly | Kurly | https://www.kurly.com/ | |
| Moneta Card | SK Telecom | http://www.moneta.co.kr/ | Merged to Foxnet in 2003 |
| Baedal Minjok | Woowa Brothers | https://www.baemin.com/ | |
| Bank Wallet | KHTC | https://www.bankwallet.co.kr/ | |
| Blue link | Hyundai Motors | http://bluelink.hyundai.com/ | |
| Beat | Beat | http://beatpacking.com/ | End Of Service in 2016 |
| Samsung Pay | Samsung Electronics | https://www.samsung.com/sec/samsung-pay/ | |
| Golfzon | Golfzon | http://www.golfzon.com/ | |
| Syrup Order | SK Planet | https://www.syrup.co.kr/ | |
| Socar | Socar | https://www.socar.kr/ | |
| Wibeetalk | Woori Bank | http://wb.briniclemobile.com/ | |
| The Pirates | The Pirates | https://www.tpirates.com/ | |
| Jeongyookgak | Jeongyookgak | https://www.jeongyookgak.com/ | |
| Kakao Bank | Kakao | https://kakaobank.com/ | |
| Kakao Taxi | Kakao | https://www.kakaocorp.com › service › KakaoT | |
| Coocha | Coocha | https://www.coocha.co.kr/ | |
| Coupang | Coupang | https://www.coupang.com/ | |
| T Money | KSCC | https://www.tmoney.co.kr/ | |
| Hwahae | Birdview | https://www.hwahae.co.kr/ | |

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
