# Peer review of "Assisting Sustainable Entrepreneurial Activities Through the Analysis of Mobile IT Services’ Success and Failure Factors"

_sustainability, doi:10.3390/su11205694_

Round 1

Reviewer 1 Report

The paper "Analyzing Success and Failure Factors Within Mobile IT services" has studed able to identify the main factors that have an influence on the success and failure of IT services based on  two identified platform models. The article contains 8 hypotheses. Verifications has carried out through using a questionnaire.
In this paper analyzed  22 mobile IT-related services that have started in South Korea as either a merchant or two-sided model. This study also reflected upon the level of the platform business maturity between a service provider and user as a way to further analyze any variations between immature and mature platforms. Next study has connected with the success or failure of IT services was measured  using a questionnaire, and  using t-tests to determine any significant  difference between the groups investigated and the three subjective evaluation factors (i.e., interests established by research hypothesis, ecosystem establishment, and technology completion) for merchant and two-sided models, which are separated from service and platform businesses.
Used also logistic regression for analysing factors of success and failure.
The article has the correct structure, references to literature are correct, however there are no references in the text to figures and tables.
Substantive remarks:
- the questionnaire was not described,
- the methods of the survey were not described,
- no information connected the survey scale and statistical analysis,
- the mothodology should be better prepare,
- it was not indicated why these models were chosen, maybe game theory would be better?

I suggest to develop an algorithm of actions for enterprises based on conclusions and implications.
After making corrections, please send me for the next review.

Reviewer 2 Report

Overall

Sustainability is of growing concern to society as a whole and exploring how business might support greater sustainability through information and communication technologies use is very welcome. The manuscript is generally well written. It presents the results of investigation into success and failures factors of mobile IT services. However, although I am of the opinion that Authors have identified an interesting and relevant area of research, this submitted paper does not do the topic sustainability. Therefore, there is a number of issues that need to be resolved before publishing in Sustainability.

Title

The title of the paper may seem an awkward: “…factors within…”. It would be probably better to use “..factors of…”.  

Introduction

The key concepts for the research should be defined in Introduction section.

First of all, it is not clear how the Authors understand “the sustainability of an enterprise” and how mobile IT services are related with the sustainability of an enterprise. This relation must be indicated to address sustainability in business that is the basic issue of Sustainability journal, whereas I found only two sentences about sustainability in this manuscript, i.e. “Thus, in order to maintain the sustainability of an enterprise  one must adequately make an analysis of the success and failure factors tailored to its own enterprise  (Introduction) in order to create a sustained competitive advantage.” (Abstract) and “Thus,  in  order  to  maintain  the  sustainability  of  an  enterprise  one  must  adequately  prepare  business with new services.” (Introduction). There are unsupported statements that need to be evidenced by the literature. There are only two statements about “sustainable business” in Introduction but this concept has not been also explained. In addition, Authors indicated “Sustainable Entrepreneurship” in the Keywords section, whereas the manuscript does not explain this word.

Secondly, mobile IT services, failure factors and success factors should be defined.

Authors should use the latest papers on ICT adoption and sustainability, e.g.  (1) L.M. Hilty & B. Aebischer, ICT for sustainability: An emerging research field, Advances in Intelligent Systems and Computing, 2015, https://doi.org/10.1007/978-3-319-09228-7_1; (2) L.W. Zacher (Ed.), Technology, society and sustainability. Selected concepts, issues and cases, Springer, https://doi.org/10.1007/978-3-319-47164-8; (3) E. Ziemba: The contribution of ICT adoption to the sustainable information society, Journal of Computer Information Systems, 2019, https://doi.org/10.1080/08874417.2017.1312635; (4) E. Ziemba, Synthetic indexes for a sustainable information society:  Measuring ICT adoption and sustainability in Polish enterprises, LNBIP 311, 2018, https://doi.org/10.1007/978-3-319-77721-4_9

In summary, I believe this topic has a great deal of value for both research and practice. However, I believe the issue lies with the current arguments as they are presented in the paper.  Authors should indicate relations between sustainability – mobile IT services – failure and success factors.

Theoretical Discussion and Hypotheses

First of all, relations between sustainability – mobile IT services – failure and success factors need to be explained in-depth based on the relevant literature.

Secondly, hypotheses should be moved from the Research methodology section here (into a relevant subsection).  The theory building subsections for the study hypotheses needs greater clarity with respect to writing. Some hypotheses are not well supported by underlying arguments (e.g. H1, H2-2, H2-3 which needs clearer arguments to support it) and/or supporting references.

Methodology

Firstly, a research instrument (a survey questionnaire) must be presented simple and clear. It is not indicated anywhere in the paper where measures were sourced. Were these measures adopted from existing studies or were they developed by Authors? If they are sources, references should be provided. If they are newly developed for this study, Authors should provide supporting arguments for the measure formulation. In addition, Authors should provide the final questionnaire items in Appendix.

Secondly, it is not clear how the constructs, i.e. Establishment of Interests, Ecosystem Establishment, Technical Completion, PC Interconnection, have been built. It should be presented which questionnaire items have been used to build them.

There is no information on what software was used to calculate statistics and why such statistics have been chosen.

Empirical Analysis  

Usage of t-test

While the t-test it is widely used for measuring the difference between groups, additional requirements should be met. Among the others there is one regarding the sample means, which stands that sample means should follow a normal distribution. Under weak assumptions it follows in large samples from the CLT (Central Limit Theorem).

Authors state that 98 cases of ICT converged were obtained from 65 ICT experts, however they were further processed by removing too old services. This process was subjective and, in that case, could provide non-representative sample. In that regard t-test could give misleading results.

There is also no assumption of variance in population. For that it could be better to provide value for modified t-test, the Welch's t-test. Authors could also provide more robust statistics methods for comparing the means of two independent samples like the bootstrapped t-test.

There is also mistake in SD of Failure specification (Table 3). The value is mistaken with t-value (19.265).

Logistic regression

Authors provided logistic regression model for all the variables (including added “dummy” variables) and then provided the general interpretation of the estimates. However interpretation could be provided in much convenient way by transforming estimates through the exp() function and obtained probability values of successes or failures. This process could greatly enhanced perception of the results by the paper reader.

It is always also worth to assess logistic regression model if it is significant or not. For that procedure authors could use likelihood ratio test of nested models.

Conclusions

Additional explanations are needed about the contribution of this paper. It should include the interpretation of the findings and describe whether the findings are in line (or not in line) with previous research presented in the literature, how the findings compare with or differ from the expected results, whether the constructed model in this paper provides the basis for constructing a new model in other areas. It needs to stress the impact of this research on sustainable business.

The implications of this study for improvement “the sustainability of an enterprise” should be indicated.

Good luck for your work on rewriting this paper!

Reviewer 3 Report

This manuscript examines the factors that lead to the success or failure of Mobile IT Services. The research context is that the advances in information technology (IT) make new mobile apps available to consumers. The authors collected surveys from 11 experienced entrepreneurs regarding 22 real IT service cases based on merchant model and two-sided model. The participants judged there were 141 success and 101 failures in total, and the t-test results confirm the distinction between success and failure of each IT service assessed by the participants was significant. Finally, the logistic regression result showed which relationships are best based on the given platform model. The findings are helpful to inform future IT service entrepreneurs and researchers involved in developing new apps based on IT services by providing a guide to what factors need to be considered before going to market.

I appreciate the effort of the authors in completing this study. However, there are not much sustainability under investigation. For the journal of Sustainability, sustainable development is the essential element, which is largely missing in this manuscript. So it is suggested that the authors submit the manuscript to another journal in the IT area.

Round 2

Reviewer 3 Report

The authors addressed the comments well, especially the connection with sustainability.